# PSEN2 Thr421Met Mutation in a Patient with Early Onset Alzheimer’s Disease

**DOI:** 10.3390/ijms232113331

**Published:** 2022-11-01

**Authors:** YoungSoon Yang, Eva Bagyinszky, Seong Soo A. An, SangYun Kim

**Affiliations:** 1Department of Neurology, Soonchunhyang University College of Medicine, Cheonan Hospital, Cheonan 31151, Korea; 2Graduate School of Environment Department of Industrial and Environmental Engineering, Gachon University, Seongnam 13120, Korea; 3Department of Bionano Technology, Gachon Medical Research Institute, Gachon University, Seongnam 13120, Korea; 4Department of Neurology, Seoul National University College of Medicine & Neurocognitive Behavior Center, Seoul National University Bundang Hospital, Seongnam 13620, Korea

**Keywords:** PSEN2, mutation, early onset AD, risk factors, risk modifiers, whole exome sequencing

## Abstract

Presenilin-2 (PSEN2) mutation Thr421Met was identified from a 57-years old patient with early onset Alzheimer’s disease (EOAD) for the first time in Korea. Previously, this mutation was discovered in an EOAD patient in Japan without a change on amyloid production from the cellular study. Both Korean and Japanese patients developed the disease in their 50s. Memory loss was prominent in both cases, but no additional clinical information was available on the Japanese patient. Magnetic resonance imaging (MRI) images of the Korean patient revealed asymmetric atrophies in both temporo-parietal lobes. In addition, amyloid positron emission tomography (PET) also revealed amyloid deposits in the gray matter of the temporo-parietal lobes asymmetrically. PSEN2 Thr421 was conserved among a majority of vertebrates (such as zebras, elephants, and giant pandas); hence, Thr421 could play an important role in its functions and any mutations could cause detrimental ramifications in its interactions. Interestingly, PSEN2 Thr421 could have homology with PSEN1 Thr440, as PSEN1 T440del mutations were reported from patients with AD or dementia with Lewy bodies. Hence, the changed amino acid from threonine to methionine of PSEN2 Thr421 could cause significant structural alterations in causing local protein dynamics, leading to its pathogenicity in EOAD. Lastly, PSEN2 Thr421Met may interact with other mutations in neurodegenerative disease related genes, which were found in the proband patient, such as ATP binding cassette subfamily A member 7 (*ABCA7*), Notch Receptor 3 (*NOTCH3*), or Leucine-rich repeat kinase 2 (LRRK2). These interactions of pathway networks among PSEN2 and other disease risk factors could be responsible for the disease phenotype through other pathways. For example, PSEN2 and ABCA7 may impact amyloid processing and reduce amyloid clearance. Interaction between PSEN2 and NOTCH3 variants may be associated with abnormal NOTCH signaling and a lower degree of neuroprotection. Along with LRRK2 variants, PSEN2 Thr421Met may impact neurodegeneration through Wnt related pathways. In the future, cellular studies of more than one mutation by CRISPR-Cas9 method along with biomarker profiles could be helpful to understand the complicated pathways.

## 1. Introduction

A majority of mutations from amyloid precursor protein (*APP*), Presenilin-1 (*PSEN1*), and Presenilin-2 (*PSEN2*) genes presented with the autosomal dominant pattern in families of patients with early onset Alzheimer’s disease (EOAD). Interestingly, over 300 mutations were reported from *PSEN1* gene on chromosome 14, while mutations in *PSEN2* on chromosome 1 were relatively rare with 87 mutations at the time of submission of this report. Structurally, *PSEN2* shared approximately 60% homology with *PSEN1* gene [1], which both contained nine transmembrane domains, connected with hydrophilic loops in cytosol and lumen and a long cytosolic loop between TM6 and TM7 [2,3].

PSEN1 and PSEN2 proteins participated as the core subunit of the γ-secretase complex for processing APP through two catalytic aspartates, the Asp263 in TM6 and the Asp366 TM7. Mutations in *PSEN2* may alter the γ-secretase cleavages of APP and Notch proteins. Elevated Aβ42/40 productions were found in four *PSEN2* mutations, including Thr122Pro, Asn144Ile, Met239Val, and Met239Ile. Cellular models of *PSEN2* Arg62His, Ser130Leu, and Val148Ile did not reveal any significant change in Aβ42/40 ratio, suggesting these mutations to be non-pathogenic or not impacting the neurodegenerations through amyloid-beta independent mechanisms [4]. Besides amyloid production, PSEN2 may influence the disease through amyloid independent pathways [5]. Mouse models with *PSEN2* Asn141Ile presented the elevated levels of oxidative stress and p53 in neurons, inducing neurodegenerations [5]. *PSEN2* knockout mice were associated with impaired cytokine productions, such as complement components, glial fibrillary acetic protein, and cathepsin S, especially in the neuroinflammatory pathways of the brain [6]. *PSEN2* mutations may also impair calcium homeostasis, where Thr122Arg or Met239Leu were associated with reduced calcium releases from intracellular compartments [7].

Clinically, *PSEN2* presented with a wide range of age of onset from 40 to 75 years of age [8]. Disease phenotypes from *PSEN2* mutations were diverse, such as late onset AD (LOAD), frontotemporal dementia (FTD), dementia with Lewy bodies (DLBs), breast cancer, and dilated cardiomyopathy (DCM) [9,10,11,12,13]. Even though the inheritance pattern of EOAD patients from PSEN2 mutations were autosomal dominant, the penetrance of mutations may be variable. Moreover, several variants were qualified as variant with unknown significance (VUS) or probably benign variants (https://www.alzforum.org/mutations/psen-2, accessed on 1 July 2022). Environmental factors were also suggested to play a significant role in AD and other neurodegenerative diseases through *PSEN2* mutations [9].

In this study, a *PSEN2* Thr421Met mutation was reported in a diagnosed patient with EOAD in Korea for the first time. Structure predictions, pathway, and biomarker analyses were performed to investigate the role of *PSEN2* Thr421Met mutation in patient with EOAD. This study was approved by the Institutional Review Board of Soonchunhyang University College of Medicine, Cheonan Hospital (IRB number: 2021-05-034-015).

## 2. Case Presentation 

### 2.1. Patient Clinical Phenotypes

The proband patient was a 56-year-old female patient, who was admitted to the hospital for complaints of gradual memory loss. She was right-handed and a retiree from an office job with a high school diploma. Her cognitive decline began 4 years before the admission in 2021, and her symptoms gradually worsened 3 years before the admission and hospitalization through the outpatient clinic. The patient repeatedly asked the same questions, such as ‘Have you been to the market?’. On the day of the hospital visit, the patient claimed that she came to the hospital by bus, even though her children drove her to the clinic. A year after the first symptom, personality changes appeared. For example, she was often angry, denying what she had done, and lost interest in her hobbies (such as farming), and she became withdrawn from family members and friends. She had difficulties in her daily activities, such as washing, bathing herself, or changing her clothes by herself. However, depression did not appear in the patient. Neurological tests did not reveal any impairment in motor and sensory functions. Her tendon reflexes were normal without any pathological reflexes, and her cerebellar function tests and gait were also normal. From the neuropsychological test, she was able to regurgitate numbers up to three digits with speaking backwards with one digit at a time. In verbal memory, impairment was observed, but the verbal fluency and comprehension were generally good. Overall, she replied to questions with short answers. Tests for apraxia and left-right orientation were normal, but the frontal lobe dysfunction (fist-hand blade-palm disorder) was observed. Spatiotemporal abilities were maintained. Based on the overall neurological diagnose, her motor and sensory were intact without facial palsy or aphasia. Her MMSE score of 10 and GDS score of 5 confirmed her diagnosis after visiting the hospital. Brain fluid-attenuated inversion recovery (FLAIR) MRI showed asymmetric atrophies in the entire brain. Uneven brain loss was most prominent in both temporo-parietal lobes. From the amyloid PET-CT data [14], intense metabolic activity could be seen in the gray matter of both temporo-parietal lobes. Interestingly, the amyloid deposits were asymmetric; they were more prominent in the left side of the temporo-parietal region, compared to the right one (Figure 1a,b). Amyloid deposits were also noticeable in these areas without clear correlations with clinical symptoms.

The final diagnosis was EOAD, and she was prescribed with donepezil, cerebrolysin, and choline alfoscerate. Later, the patient was discharged from the hospital with a stable condition and recommended for the continuous medications and regular visits through the outpatient clinic. The patient had hypertension as an underlying disease, but she did not drink or smoke. The concentrations of cerebrospinal fluid (CSF) Aβ42 was quantified using commercial ELISA kits (INNOTEST β-AMYLOID (1–42), according to the manufacturer’s instructions. No significant reduction was observed in CSF-Aβ42 (906.3 pg/mL), compared to the normal controls (941.5–1238.2 pg/mL). Western blot for 14–3-3 protein, RT-QUIC for PrP^Sc^ and Tau quantification was performed by Korea Creutzfeldt-Jakob Disease (CJD) Diagnostic Center and Hallym University Medical Center (Seoul, Republic of Korea). The patient was negative for both 14–3-3 protein, and abnormal prion protein, which ruled out the possibility of sporadic CJD diagnosis. Total Tau levels were increased (379.9 p/mL) in the patient, compared to the healthy controls (below 200 pg/mL). Family history for any neurodegenerative diseases seemed to be negative, and no other affected family member was found. Segregation of mutation could not be proven since all relatives refused the genetic testing. Apolipoprotein E (*APOE*) genotype from whole exome sequencing (WES) and Sanger sequencing revealed E3/3 genotype.

### 2.2. Genetic Analysis

Whole blood sample from the patient was received, and DNA was extracted from white blood cells by Quiagen blood kit (Seoul, Korea). The WES analysis was performed with the patient’s genomic DNA sequencing results with Illumina platform by Novogene Inc. (https://en.novogene.com; accessed on 1 July 2021). The sequencing data was visualized by Integrative Genomics Viewer (IGV) software [15]. Sanger sequencing was performed to verify the mutation. Other causative mutations from other responsible genes for the neurodegenerative diseases, including AD risk factors, Parkinson’s disease, frontotemporal disease, amyotrophic lateral sclerosis, and vascular diseases, were searched (Appendix A). STRING and Cluego pathway analyses were also performed on the mutations from the patient to estimate its associations with other genetic factors [16].

PSEN2 Thr421Met mutation was analyzed by in silico bioinformatic tools, such as PolyPhen-2 (http://genetics.bwh.harvard.edu/pph2/, accessed on 1 July 2022), SIFT (http://sift.jcvi.org/, accessed on 1 July 2022), PROVEAN, and CADD (https://cadd.gs.washington.edu/, accessed on 1 July 2022) tools. The 3D protein structure predictions were performed on normal PSEN2 421Thr and 421Met mutation by Phyre2 tool (http://www.sbg.bio.ic.ac.uk/phyre2/html/page.cgi?id=index, accessed on 1 July 2022). The structural alterations between PSEN2 421Thr and 421Met were visualized by the Discovery Studio 3.5 Visualizer tool.

## 3. Results

The patient carried a previously reported *PSEN2* mutation, c.1259C>T or p.Thr421Met (g.226895494.C>T, Figure 2), from Japan. *PSEN2* Thr421Met was not updated in the 1000Genomes database (http://www.internationalgenome.org/; accessed on 1 July 2022), but it appeared among the GnomAD database (https://gnomad.broadinstitute.org/; accessed on 1 July 2022) with the allover frequency of 0.00002788. Thr421Met was found in seven unaffected individuals from GnomAD, five Asians and two non-Finnish European individuals. Among them, age range was known among five individuals: two of them were between 45 and 50 and 60 and 65, while one other carrier was between 65 and 70 years of age.

Along with *PSEN2* Thr421Met, additional common or rare variants appeared in the patient in a different neurodegenerative disease. Among AD risk genes, ATP Binding Cassette Subfamily A Member 7 (*ABCA7*) bound three relatively rare variants (Thr319Ala, His395Arg, and Arg463His). Furthermore, *ABCA7* carried a common AD risk factor variant, Gly1527Ala [17]. Additional variants were found in Cas Scaffold Protein Family Member 4 (*CASS4*, Pro660Ser), Sialic acid binding Ig-like lectin 3 (*CD33*, Ala14Val), Ephrin Type-A Receptor A1 (*EPHA1*; Met900Val and Val160Ala), Solute Carrier Family 24 Member 4 (*SLC24A4*; Lys533Gln), and Sortilin Related Receptor 1 (*SORL1*; Gln1074Glu and Val1967Ile). Among the non-AD risk genes, such as Notch Receptor 3 (*NOTCH3*), Leucine-rich repeat kinase 2 (*LRRK2*), and microtubule-associated protein tau (*MAPT*), several variants appeared, too. A rare mutation, Leu1518Met, was observed in *NOTCH3*, but its role in neurodegenerative diseases remained unclear [18]. Additionally, a common variant was observed in *MAPT* (Tyr441His), and three common variants appeared in *LRRK2* (Arg50His, Ser1647Thr, and Met2397Thr) (Appendix A). STRING pathway analysis (Figure 3a) suggested that PSEN2 could directly interact with other AD risk genes, such as *ABCA7*, *SORL1*, *CD33*, *CASS4*, and *SLC24A4*. STRING networking also revealed direct interactions with non-AD risk genes, such as *MAPT*, *NOTCH3,* and *LRRK2*. ClueGo analysis confirmed the *PSEN2* association with AD risk genes, such as *ABCA7* and *SORL1* in amyloid processing, transport, and amyloid clearance. An additional direct association was found between PSEN2, Huntingtin Interacting Protein 1 Related (*HIP1R*), and low-density lipoprotein receptor-related protein 6 (*LRP6*), which could impact the protein transport. Additionally, PSEN2 could impact negative regulation of endocytosis through ABCA7 interactions by interacting ataxin-2 (*ATXN2*). In addition, between PSEN2 could play a role in lysosomal transport through SORL1 by interacting with different genes, such as *MAPT*, *LRRK2*, Senataxin (*SETX*), Cyclin G Associated Kinase (*GAK*), or VPS11 Core Subunit Of CORVET And HOPS Complexes (*VPS11*). Furthermore, between *PSEN2* and Sacsin Molecular Chaperone (*SACS*) genes could interact indirectly through *SORL1* and impact the protein aggregations. Indirect associations between *PSEN2* and ATXN2 were also found through *ABCA7*, which may be related to negative regulation of endocytosis (Figure 3b, Appendix A).

The mutation was predicted as a damaging variant by PolyPhen2 and SIFT analyses with the score of 1.0 and 0.01, respectively. CADD prediction revealed relatively high scores (25.8), suggesting Thr421Met as a damaging variant. Multiple sequence alignment for PSEN2 Thr421 revealed that the majority of vertebrate species (zebras, elephants, and giant pandas) carried threonine at the homologous position in their presenilin-like sequences. However, phenylalanine was at Thr421 position in Nile tilapia and Japanese pufferfish. Alanine was found at homologous residue in purple sea urchin. No methionine at the homologous position was found in any animal species. Taken together, Thr421 was a conserved residue among all vertebrates, and any mutation at Thr421 could cause detrimental effects in its function. Structure predictions revealed that Thr421Met may result in disturbances of protein structure or in intramolecular interactions. Normally, Thr421 would form double hydrogen bonds with Leu424 and Pro417 and a single hydrogen bond with Ile418. In the case of Met421, all contacts remained, but one hydrogen bond may be lost with Pro417. This may result in extra stress and structural alterations inside the helix, causing abnormal motion of TM9. Threonine and methionine are polar and non-polar residues, respectively. Furthermore, the larger size of methionine could result in additional conformational stress inside the PSEN2 helix through the increased hydrophobicity. A lost hydroxyl group in the case of methionine may disturb the interactions between PSEN2 and its potential binding partner (Figure 4).

## 4. Discussion

*PSEN2* Thr421Met mutation was initially found in a Japanese EOAD patient. Limited information is available on clinical phenotype associated with the mutation. Previously, the Japanese patient developed memory impairment at the age of 55 and carried the homozygous *APOE* E4 allele. Inheritance pattern of the mutation remained unclear, since no sample could be obtained from any relative of the patient. No further details were described on the Japanese patient’s disease phenotype nor brain imaging data. The same Japanese study revealed that *PSEN2* Thr421Met was not found in 112 unaffected Japanese individuals and in 147 non-AD patients. They suggested that *PSEN2* Thr421Met could play an important role in disease process by interacting with other disease risk factors, such as homozygous *APOE* E4 allele [19]. No additional studies were found, which described the clinical phenotypes of AD patients with PSEN2 Thr421Met mutation.

The Korean patient with *PSEN2* Thr421Met developed memory loss and personality changes at the age of 55. MRI of our patient revealed the asymmetric neural losses and atrophies in the temporo and parietal lobes, which were not typical in EOAD patients. The images from amyloid PET also revealed the asymmetric amyloid deposits in the gray matter of the temporo-parietal lobes, especially in the left lobes. Even though strong amyloid signal was seen in amyloid PET (and elevated levels of CSF-total Tau), the ELISA for CSF- Aβ42 did not reveal significant reduction. Inconsistence in amyloid PET and CSF-Aβ42 levels may be possible. CSF-amyloid positivity may be possible in normal controls (while they are negative for amyloid PET), suggesting that CSF-amyloids may be a more sensitive marker in pre-clinical or early disease stages, while amyloid PET may be a strong marker for more advanced disease stage [20]. Additionally, amyloid PET may be a stronger marker for AD diagnosis, compared to CSF Aβ42 in the case of differential diagnosis of neurodegenerative diseases [21]. AD patients were reported before, who were positive to amyloid PET and CSF-Tau, but no reduction was observed in CSF-amyloid levels. One of the explanations could be that CSF-amyloid only reflects the soluble pool of amyloid peptides, while amyloid PET may represent the fibrillar amyloid aggregations. Moreover, CSF Aβ42 and amyloid PET could be possibly related independently and differentially to the other, non-amyloid types of AD pathology (such as CSF-Tau, hippocampal atrophy, APOE E4 allele, or reduced cerebral blood flow) [22,23]. CSF amyloid may be strongly correlated with *APOE* E4 allele, while PET positivity may be associated with CSF-Tau positivity or Alzheimer’s Disease Assessment Scale-cognitive subscale (ADAS-cog). The strong relationship between Tau positivity and ADAS-cog also confirmed that amyloid PET may be a more effective diagnostic marker in later disease stages, compared to CSF-amyloid. PET-positivity and CSF Aβ42 negativity may also suggest that fibrillar amyloid or short amyloids, such as Aβ40, should also be considered as potential CSF markers [20,21,22,23]. Since our patient had *APOE* E3/3 genotype, other genetic risk factors may contribute strongly to the disease onset. Table 1 presents the comparison of the Korean and Japanese patients with *PSEN2* Thr421Met mutation.

The second report, which mentioned *PSEN2* Thr421Met mutation, investigated the genetic background of hidradenitis suppurativa (HS), which is a chronic inflammatory disease [24,25]. HS results from the abnormal and uncontrolled inflammatory processes in the cutaneous follicular unit, leading to pain in the sinus, abscess, and hypertrophic scarring. One Japanese patient with the *PSEN2* Thr421Met was positive for HS, causing splice site mutation (c.582+1delG) in the nicastrin (*NSCTN*) gene. Affected individuals presented a scarring and fistulae, which were mostly located at the neck and perineal regions. Unfortunately, this study failed to perform phenotype-genotype correlation in the patient’s family, since all of the relatives refused the genetic testing. This study suggested that *PSEN2* Thr421Met may not be a main causative factor for HS. Additionally, this study did not mention whether the HS patient had any neurodegenerative disease phenotype [24,25].

*PSEN2* Thr421 shared homology with Thr440 in *PSEN1* (Table 2). The patient with *PSEN1* Th440 deletion developed a rapid progressive AD with Lewy bodies and cotton wool plaques [26]. Deletion of Thr440 could reduce the Aβ42 levels and fully eliminate the Aβ40 productions [27]. However, a cell study suggested that *PSEN2* Thr421Met may be a non-damaging variant. Hsu et al. (2020) cloned *PSEN2* Thr421Met into neuroblastoma cells without affecting the γ-secretase activity. The mutation reduced both Aβ42 and Aβ40 levels without the change in the Aβ42/Aβ40 ratio [28].

Even though *PSEN2* Thr421Met was suggested as a possible benign variant, it may be fully ruled out for its influence in the disease progression, especially when Thr421 residue was conserved among vertebrates and even between *PSEN1* and *PSEN2*. This mutation received relatively high CADD scores (25.8), suggesting the damaging effects. Structure prediction showed that Met421 may alter the interactions with Pro417 by losing a hydrogen bond. Pro417 was part of the PALP (Pro414-Ala415-Leu416-Pro417) motif in *PSEN2.* The PALP motif in PSENs played a critical role in catalytic site formation and APP recognition by γ-secretase. Disturbance of intramolecular interactions of PALP residues and within any nearby residues may disturb the γ-secretase functions [29].

Mutant *PSEN2* may interact with other mutant AD or non-AD risk factor genes (Figure 3a,b, Appendix A). Presenilins may have multiple roles in cell functions by interacting with several proteins. It may be possible that variants in these genes could play a role in disease progression in the presence of *PSEN2* Thr421Met [30]. Pathway analyses by ClueGo and STRING revealed that *PSEN2* could interact with *ABCA7* and *SORL1. ABCA7* was originally involved in lipid metabolisms with its important role in the amyloid metabolism. *PSEN2* mutations may impact amyloid pathology by altering the APP trafficking and availability of APP cleavage products [31]. *ABCA7* deficiency revealed to increase the amyloid productions and accumulations through increasing β-secretase activity or reducing the amyloid clearance mechanisms [32]. It could be possible that *ABCA7* and *PSEN2* could interact and share the common pathways through amyloid-β processing [33]. *SORL1* dysfunctions may also be important in AD pathogenesis. *SORL1* may play an important role in endosomal degeneration and neuronal recycling process [34,35]. STRING revealed that *PSEN2* may share common pathways with other AD risk genes, such as *CASS4, EPHA1,* and *SLC24A4*, but no detailed reports are available through which mechanisms they may interact. *EPHA1* was verified to play a role in axon guidance, synaptic development, and plasticity. *CD33* was verified to impact cell communication and also in the regulation of adaptive immune system. [36,37]. *CASS4* may impact the cytoskeletal functions and also APP metabolism [37] *SLC24A4* was verified as an ion transporter, and its dysfunctions may result in defects in calcium ion transport, leading to neurodegeneration [38]. ClueGo networking revealed a direct association between *LRP6* and *PSEN2*. *LRP6* is an AD risk factor, which may play a role in the endocytosis of lipoproteins and their ligands. Deficiency of *LRP6* may significantly impact AD risk through Wnt signaling. Additionally, mutant *LRP6* was suggested to increase amyloid processing and Aβ42 aggregation [39].

Among the non-AD related genes, STRING analysis supported the interactions between *NOTCH3* and *PSEN2*. As part of γ-secretase, *PSEN2* (as well as *PSEN1*) could impact the cleavage of other substrates, including Notch components [9]. Even though mutations in *NOTCH3* were verified as causative factors for CADASIL, they may also impact AD pathology [40]. *PSEN2* mutations, such as Arg62His and Arg71Trp, may destabilize the PSEN2 protein and impair the Notch signaling. In *Caenorhabditis elegans* and mouse fibroblast models, *PSEN2* variants could result in the reduced Egl rescue in comparison to wild type [13]. Several AD causing mutations in *PSEN1* may also reduce the cleavage of Notch proteins, such as Leu166Pro [41], Leu392Val [42], Cys410Tyr [43], Leu435Phe, and Pro436Ser [44]. Notch signaling played a crucial role in brain development, and its impairment could impact several neurodegenerative diseases significantly, including AD and PD. Notch impairment reduced the amyloid clearance, neuroprotection, or the self-renewal of nerve cells through creating disturbances in microtubule stability [45]. STRING indicated the direct associations between LRRK2 and PSEN2, since they both shared a common pathway through the Wnt signaling, such as beta catenin or low-density lipoprotein receptor-related protein 6 (LRP6). Hence, LRRK2 may serve as a bridge between the membrane and cytosolic components of Wnt proteins. Impairments in LRRK2 functions may result in reduced levels of Wnt signals, enhancing the neurodegenerations [46]. PSEN1 or PSEN2 mutations could also impact Wnt signaling. PSENs may control beta catenin stability, and their impaired or altered interactions could reduce the degree of Wnt signals [47]. ClueGo revealed a direct association between *PSEN2* and *HIP1R* gene through receptor cargo activity. Association of *HIP1R* to AD or *PSEN2* was not confirmed. However, *HIP1R* may impact signal transduction and control the transport of growth factor receptors, so the possible common mechanisms between *PSEN2* and *HIP1R* may not be ruled out [48]. A limitation of this study is that these interactions could not be confirmed in vitro. Cellular studies will be carried out in the future in the presence and absence of potential AD risk modifiers. Furthermore, since the patient’s relatives refused the genetic test or giving any detailed information on their health status, the segregation analysis on *PSEN2* Thr421Met of the Korean family could not be performed.

## 5. Conclusions

In summary, the significance of *PSEN2* Thr421Met in neurodegeneration may not be overlooked. The mutation may interact with other genetic AD and non-AD risk factors, impacting the multiple disease related pathways, especially with *ABCA7* for influencing the amyloid processing pathway. Furthermore, the amyloid-independent pathway in the neurodegeneration through the impaired Notch or Wnt signaling pathways should be investigated.

## Figures and Tables

**Figure 1 ijms-23-13331-f001:**
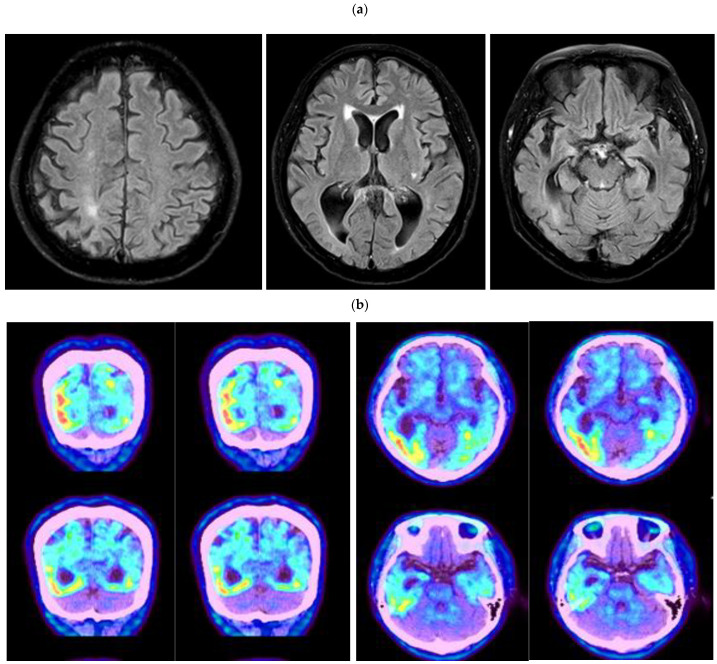
(**a**) Brain MRI of the patient, revealing asymmetric brain atrophies for the entire brain. Brain shrinkage was most prominent in the temporo-parietal lobe. (**b**) Amyloid PET-CT revealed intense amyloid deposits in the gray matter area asymmetrically. Amyloid deposits were prominent in both parietal and temporal lobes, but they were more prominent in the left temporo-parietal lobes.

**Figure 2 ijms-23-13331-f002:**
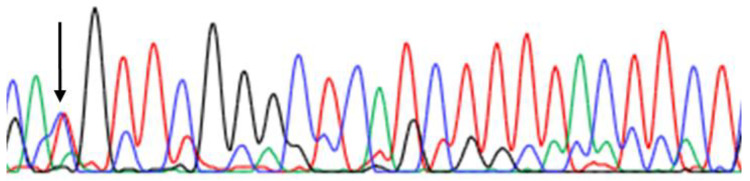
Sanger sequencing data of heterozygous PSEN2 Thr421Met mutation, indicated with arrow.

**Figure 3 ijms-23-13331-f003:**
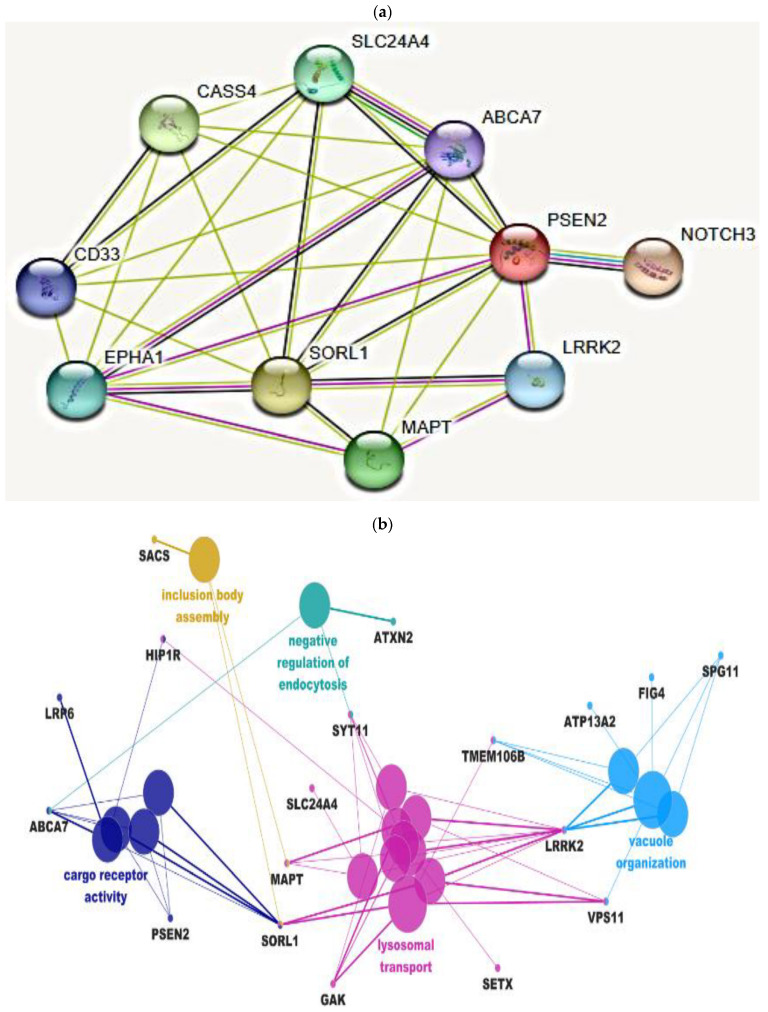
(**a**) STRING pathway analysis on patient with *PSEN2* Thr421Met mutation: *PSEN2* may interact directly with different AD (such as *EPHA1, ABCA7,* and *SORL1*) and non-AD (*NOTCH3, MAPT,* and *LRRK2*) risk genes. (**b**) Pathway analysis by ClueGo confirmed that *PSEN2* may interact with *SORL1* and *ABCA7* genes, involved in cargo receptor activity. Further indirect associations also appeared between *PSEN2* and *LRRK2, GAK, SETX,* or *SLC24A4*, which may impact lysosomal transport.

**Figure 4 ijms-23-13331-f004:**
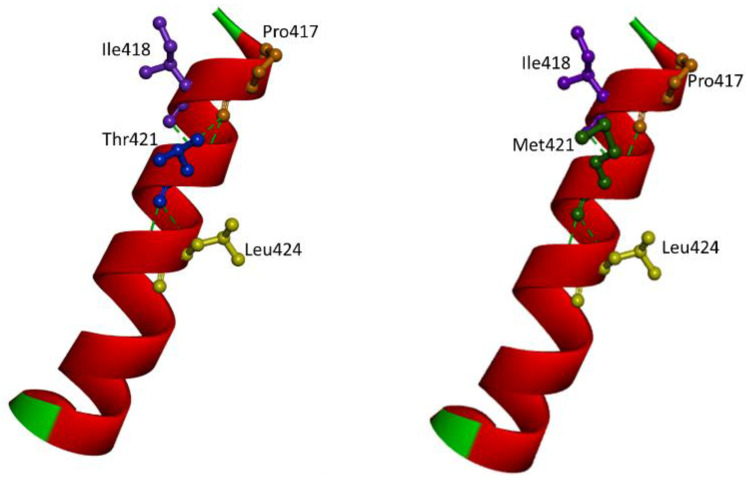
Intramolecular interactions in case of Thr421 and Met421. Mutation may result in a loss of a hydrogen bond with Pro417.

**Table 1 ijms-23-13331-t001:** Comparison of *PSEN2* Thr421Met cases from Japan and the newly discovered Korean case.

	Korean Patient	Japanese Patient
*Disease*	EOAD	EOAD
*Age of onset*	55	55
*Gender*	F	F
*Family history*	negative	Unclear, probable sporadic
*APOE*	3/3	4/4
*Imaging*	MRI: asymmetric atrophies of both temporo-parietal lobes; Amyloid PET-CT, asymmetric amyloid deposits in gray matter of temporo-parietal lobes.	NA
*Clinical symptoms*	Memory loss, personality changes	Memory decline

**Table 2 ijms-23-13331-t002:** Comparison of *PSEN1* Thr440del and *PSEN2* Thr421Met.

	*PSEN2* Thr421Met	*PSEN1* Thr440del
Disease	EOAD	EOAD/DLB
Symptoms	Typical EOAD, initial symptom was amnesia, personality changes	Parkinsonism, cognitive decline, generalized dystonia
Age of onset	55	52
Family history	unclear	Probable positive
Imaging	Atrophy in temporo-parietal lobes, PET revealed asymmetric plaques in the same areas	Neuronal loss in different brain areas: substantia nigra and cerebral cortex
Brain hallmark	Amyloid plaques in in gray matter of temporo-parietal lobes	Cotton wool plaques, cerebral amyloid angiopathy
Functional studies	Reduced Aβ42 and 40 levels, no effect on Aβ42/40 ratio	Reduced Aβ42 levels, abrogated Aβ40 production

## Data Availability

Not applicable.

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
