# Peer review of "PSEN2 Thr421Met Mutation in a Patient with Early Onset Alzheimer’s Disease"

_ijms, 2022, doi:10.3390/ijms232113331_

Round 1
Reviewer 1 Report
In this manuscript, authors describe the first case in Korea of a PSEN2 mutation at Thr421. They describe asymmetric atrophies in the brain and an accumulation of amyloid deposits. By exome sequencing, they describe different rare variants and possible interactions with different genes related to AD.
Minor:
-Authors should revise the use of verbs in “past tense” and the grammar in some paragraphs (lines 247, 261-263..)
-Fig 3 and 4. These figures could improve with a better resolution, and particularly for fig. 3, letter size should be bigger.
- This is the first case of a PSEN2 Thr421Met individual affected by AD in Korea, but are there other cases reported in the rest of the world, apart from that in Japan?
Major:
-I wonder if authors could get blood or CSF samples from the patient. It would be very interesting to measure AD biomarkers abeta, tau, fosfo-tau, to increase the amount of data from this patient.
-The paper is a bit excessive in suggesting/hypothesis of networks/connexions from the data obtained from the genome studies. Discussion of genes networks is a bit messy also. I think the manuscript would improve whether the authors organize better the rationale of their findings.
Author Response
In this manuscript, authors describe the first case in Korea of a PSEN2 mutation at Thr421. They describe asymmetric atrophies in the brain and an accumulation of amyloid deposits. By exome sequencing, they describe different rare variants and possible interactions with different genes related to AD.
Thank you very much for the comments, we tried to revise it according to your suggestions.
Minor:
-Authors should revise the use of verbs in “past tense” and the grammar in some paragraphs (lines 247, 261-263..)
Thank you, grammar has been improved.
-Fig 3 and 4. These figures could improve with a better resolution, and particularly for fig. 3, letter size should be bigger.
Thank you, the issue has been fixed.
- This is the first case of a PSEN2 Thr421Met individual affected by AD in Korea, but are there other cases reported in the rest of the world, apart from that in Japan?
To date, no report was found, which mentioned this mutation in EOAD cases, other than the Japanese case.
Major:
-I wonder if authors could get blood or CSF samples from the patient. It would be very interesting to measure AD biomarkers abeta, tau, fosfo-tau, to increase the amount of data from this patient.
Thank you for the comment. Currently, we are trying to collect additional samples for biomarker studies. If we successfully obtain the data from the patient’s CSF, we will prepare an addendum to the manuscript.
-The paper is a bit excessive in suggesting/hypothesis of networks/connexions from the data obtained from the genome studies. Discussion of genes networks is a bit messy also. I think the manuscript would improve whether the authors organize better the rationale of their findings.
Thank you for the suggestion. We tried to re-organize the pathway data. We added several additional information, based on the possible common pathways between PSEN2 and other risk genes. We also tried to separate the association of PSEN2 with AD risk genes and PSEN2 with non-AD risk genes. We added tables in the supplement file on genes, which may be related directly or indirectly with PSEN2
Reviewer 2 Report
The authors have done wonderful work and it is really easy and straightforward to recommend this manuscript for publication. First I like that the authors have described the mutation at the clinical level, and also at the protein structure level. A such span of interest is not frequently seen. Second I like that the authors have given credit to the earlier studies on Japanese patients, and clearly indicated the new insights that their study brings.
Tables 1 and 2 give a really nice summary.
Of course, someone can criticize the study because it lacks CSF Abeta x-40 or x-42 measurements. However, as a biochemist, I would argue that those measurements are frequently performed sloppy, and incomplete. I respect the author’s decision to leave different Abeta x-40 or x-42 measurements for some future studies that can have the required depth. I am personally looking forward to incorporating this mutation in my studies of changes in g-secretase structure.
I recommend this manuscript for publication. Congratulations to the authors.
Author Response
The authors have done wonderful work and it is really easy and straightforward to recommend this manuscript for publication. First I like that the authors have described the mutation at the clinical level, and also at the protein structure level. A such span of interest is not frequently seen. Second I like that the authors have given credit to the earlier studies on Japanese patients, and clearly indicated the new insights that their study brings.
Tables 1 and 2 give a really nice summary.
Of course, someone can criticize the study because it lacks CSF Abeta x-40 or x-42 measurements. However, as a biochemist, I would argue that those measurements are frequently performed sloppy, and incomplete. I respect the author’s decision to leave different Abeta x-40 or x-42 measurements for some future studies that can have the required depth. I am personally looking forward to incorporating this mutation in my studies of changes in g-secretase structure.
I recommend this manuscript for publication. Congratulations to the authors.
Thank you very much for the positive review, we really appreciate it. Currently, we are trying to collect additional samples for biomarker studies. If we successfully obtain the data from the patient’s CSF, we will prepare an addendum to the manuscript.
Round 2
Reviewer 1 Report
Authors have included my suggestions , so I think the paper has improved. However, I suggest to include the data from the patient’s CSF in the manuscript, instead of in the addendum. It will be very interesting to match these new data with those already in the manuscript, otherwise, this work still lacks of some strenght to characterize this mutation.
Author Response
Authors have included my suggestions , so I think the paper has improved. However, I suggest to include the data from the patient’s CSF in the manuscript, instead of in the addendum. It will be very interesting to match these new data with those already in the manuscript, otherwise, this work still lacks of some strength to characterize this mutation.
Thank you very much for the constructive comment. We analyzed the patient’s CSF and added new paragraphs in the Case description and discussion.
Materials and Methods, Patient clinical phenotypes
“The concentrations of CSF Aβ42 and T-tau were quantified using commercial ELISA kits (INNOTEST β-AMYLOID(1–42) according to the manufacturer’s instructions. No significant reduction was observed in CSF-Ab42 (906.3pg/mL), compared to the normal controls (941.5-1238.2 pg/mL). Western blot for 14-3-3 protein and RT-QUIC for PrPSc was performed by Korea CJD Diagnostic Center and Hallym University Medical Center (Seoul, Republic of Korea). Patient was negative for both 14-3-3 protein, and abnormal prion protein, which ruled out the possibility of sporadic CJD diagnosis. Total Tau levels were increased (379.9p/mL) in the patient, compared to the healthy controls (below 200 pg/mL). “
Discussion
‘Even though strong amyloid signal was seen in amyloid-PET (and elevated levels of CSF-total Tau), the ELISA for CSF- Ab42 did not reveal significant reduction. Inconsistency in amyloid PET and CSF-Ab42 levels may be possible. CSF-amyloid positivity may be possible in normal controls (while they are negative for amyloid-PET), suggesting that CSF-amyloids may be a more sensitive marker in pre-clinical or early disease stages, while amyloid PET may be strong marker for more advanced disease stage [PMID: 25541191]. Also, amyloid-PET may be stronger marker for AD diagnosis, compared to CSF Ab42 in case of differential diagnosis of neurodegenerative diseases [PMID: 33466854]. AD patients were reported before, who were positive to amyloid-PET and CSF-Tau, but no reduction was observed in CSF-amyloid levels. One of the explanations could be that CSF-amyloid only reflects the soluble pool of amyloid peptides, while amyloid-PET may represent the fibrillar amyloid aggregations. Also, CSF Ab42 and amyloid PET could be possibly related independently and differentially to the other, non-amyloid types of AD pathology (such as CSF-Tau, hippocampal atrophy, APOE E4 allele or reduced cerebral blood flow) [PMID: 26468410, PMID: 26468410, PMID: 25541191]. CSF amyloid may be strongly correlated with APOE E4 allele, while PET positivity may be associated with CSF-Tau positivity or Alzheimer's Disease Assessment Scale-cognitive subscale (ADAS-cog). The strong relationship between Tau positivity and ADAS-cog also confirmed that amyloid PET may be more effective diagnostic marker in later disease stages, compared to CSF-amyloid. PET-positivity and CSF Ab42 negativity may also suggest that fibrillar amyloid or short amyloids, such as Ab40 should also be considered as potential CSF markers [PMID: 25541191, PMID: 31464088]. ‘
Round 3
Reviewer 1 Report
Authors have added CSF information as requested. I think the new data improve the quality of paper